# Speed Up Iterative Non-Autoregressive Transformers by Distilling Multiple Steps

## Abstract

The computational benefits of iterative non-autoregressive transformers decrease as the number of decoding steps increases. As a remedy, we introduce **Di**still **M**ultiple **S**teps (**DiMS**), a simple yet effective distillation technique to decrease the number of required steps to reach a certain translation quality. The distilled model enjoys the computational benefits of early iterations while preserving the enhancements from several iterative steps. DiMS relies on two models namely *student* and *teacher*. The student is optimized to predict the output of the teacher after multiple decoding steps while the teacher follows the student via a slow-moving average. The moving average keeps the teacher's knowledge updated and enhances the quality of the labels provided by the teacher. During inference, the student is used for translation and no additional computation is added. We verify the effectiveness of DiMS on various models obtaining 7 and 12.9 BLEU points improvements on distilled and raw versions of WMT'14 De-En, respectively.

## 1 Introduction

Neural machine translation models typically follow an autoregressive decoding strategy, generating the target sentence one token at a time. This sequential nature makes the inference process slow and dependent on the output sequence length. To address this limitation Gu et al. (2018) introduces the Non-Autoregressive Transformer (NAT). NAT generates the entire target sentence in parallel, reducing the latency by an order of magnitude. NAT can be considered as a member of a broader family of iterative non-autoregressive Transformers (iNAT) (Lee et al., 2020; Stern et al., 2019; Ghazvininejad et al., 2019) where the number of decoding steps is fixed and independent of the sequence length. By tuning the number of decoding steps, one can control the trade-off between speed and quality. While iNATs can be considered as efficient alternatives to their autoregressive counterparts, Kasai et al. (2020b) shows that autoregressive models can be sped up without loss in accuracy by combining shallow decoders with deep

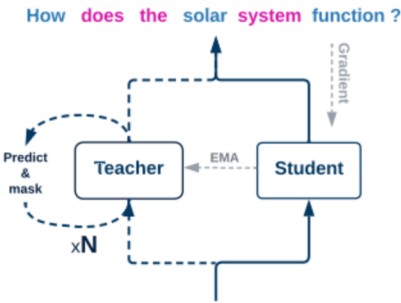

Figure 1: DiMS training. The student is trained to match the predictions of the teacher after several iterative steps. Teacher is updated with an exponential moving average of the student.

encoders. This diminishes the computational advantage of iNATs and challenges their motivation. The focus of recent work has thus shifted to design single-step NAT models (Ghazvininejad et al., 2020a; Qian et al., 2021; Du et al., 2021).

In order to preserve the enhancements obtained by multiple decoding iterations of iNATs, we introduce **Di**still **M**ultiple **S**teps (**DiMS**), a distillation algorithm applicable to a wide range of iterative models. Given a pre-trained iNAT, referred to as *teacher*, a *student* aims to replicate the behavior of multiple iterative steps of the teacher with one decoding pass. This process resembles the well-known knowledge distillation framework (Hinton et al., 2015). However, instead of reducing the number of parameters, we aim to decrease the number of decoding passes. The final model then enjoys the translation quality of multi-steps iNAT with the computational efficiency of single-step translation.

The proposed distillation can be repeated iteratively, where at the end of each round the newly optimized student becomes the next teacher. While effective, iterative distillation is slow as it requires multiple rounds of training until convergence. Alternatively, we propose updating the parameters of the teacher with an exponential moving average (EMA) of the student. This gradually transfers the new knowledge learned by the student to the teacher and can be viewed as a continuous variant of iterative distillation. Figure 1 depicts the DiMS algorithm on a high level.

We demonstrate the effectiveness of our approach on several public datasets by showing that DiMS obtains substantial improvements on single-step translation with gains of up to 7 BLEU points on the distilled training dataset, while the gains on raw datasets are even greater. Notably, we are able to surpass many leading NAT models designed specifically for single-step translation and we set a new state of the art on raw datasets. We further show that EMA considerably speeds up training and converges to a comparable accuracy with iterative distillation in a fraction of epochs.

## 2 BACKGROUND

In this section, we lay out a formal framework for iNATs. We use the setup of Conditional Masked Language Models (CMLM), the approach first introduced in Ghazvininejad et al. (2019) and subsequently adopted in many iNAT models (Ghazvininejad et al., 2020b; Kasai et al., 2020a; Saharia et al., 2020; Huang et al., 2021). The source sentence, target sentence, and target sequence length are denoted by $\mathbf{x}$, $\mathbf{y}$ and $N$, respectively.

### 2.1 TRAINING

Given a partially masked reference sentence $\tilde{\mathbf{y}}$ and the corresponding source context $\mathbf{x}$, the model is trained to reveal all the masked positions simultaneously (Ghazvininejad et al., 2019). From a probabilistic perspective, this imposes a conditional independence assumption on the predicted tokens. Formally, the training loss is:

$$\mathbb{E}_{\tilde{\mathbf{y}} \sim \mathbf{M}(\mathbf{y})} \sum_{i \in \xi(\tilde{\mathbf{y}})} - \log p_\theta(y_i | \mathbf{x}, \tilde{\mathbf{y}}),$$

where $\mathbf{M}$ is a distribution over all partially masked target sentences and $\xi$ is a function that returns the set of masked indices. The training objective above implicitly assumes access to the target sentence length. To resolve this issue, CMLM trains a parametric model, *length predictor*, to predict the output length.

### 2.2 INFERENCE

The inference begins by creating a template $\tilde{\mathbf{y}}^{(0)}$ with $\tilde{N}$ masked tokens, where $\tilde{N}$ is the output of the length predictor. At iteration $t$ of the inference, the model predicts the translation $\mathbf{r}^{(t)}$ given $\tilde{\mathbf{y}}^{(t-1)}$ and $\mathbf{x}$ as inputs. Depending on the number of decoding iterations $S$, typically a linear unmasking policy is used where at each step $\tilde{N}/S$ tokens with the highest probability are revealed. This process is repeated $S$ times, resulting in a fully revealed sentence. Denoting the output probability of the model by $p_\theta$ , the $t$-th step of the inference can be formally written as:

$$\tilde{y}_i^{(t)} = r_i^{(t)} \ \text{ if } \ i \in \underset{k=\frac{\tilde{N}}{S}}{\text{arg-topk}} \ \left\{ p_\theta \left( r_j^{(t)} \Big| \mathbf{x}, \tilde{\mathbf{y}}^{(t-1)} \right) \right\}_{j \in \xi(\tilde{\mathbf{y}}^{(t-1)})} \ \text{ else } \ \tilde{y}_i^{(t-1)}.$$

Note that multiple length candidates can be considered (e.g. $\tilde{N} \pm 1$) with the average token probability as a ranking criterion. This is similar to beam search in autoregressive models but applied to the output sequence length. It is referred to as *length beam*.

## 3 DISTILLATION OF ITERATIVE NON-AUTOREGRESSIVE TRANSFORMERS

Increasing the number of decoding steps diminishes the computational advantage of iNATs. Our objective is to limit the number of decoding steps without degrading the performance. More specifically, we want to compress the performance gain from multiple iterative steps of a teacher into one decoding

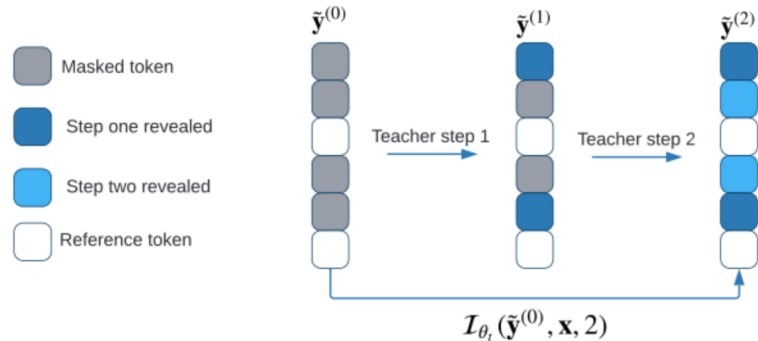

Figure 2: Two iterative steps of the teacher applied to a partially masked sentence.

pass of a student. For instance, consider an iterative model (teacher) that uses eight decoding steps. By replicating four steps of the teacher with one decoding pass, two steps of the student would be sufficient to reach a similar performance. Ideally, one would hope for a single iteration of the student to replicate the top performance of the teacher, but the model's limited expressiveness might prevent this.

The standard way of knowledge distillation would have the teacher generate soft labels for all intermediate iterations, and optimize the student to track the teacher's output with fewer steps, but doing such generation on-the-fly greatly increases the training cost. One can move this process to a pre-processing phase, at the cost of large memory requirements. We propose to use partially masked reference sentences as an approximation to the intermediate predictions of the teacher, which eliminates the need for several decoding passes or large memory capacity.

The distillation process starts by initializing the student and the teacher to the same pre-trained model with parameters $\phi$ i.e. $\theta_s = \theta_t = \phi$ where $\theta_s$ and $\theta_t$ denote the parameters of the student and teacher respectively. Then, the teacher processes a partially masked sentence $\tilde{y}$ through $n$ iterative steps with a linear unmasking policy. More precisely, $i/n$ of the originally masked tokens are revealed up to step $i$ and after the final pass, no masked token remains. This is similar to the inference procedure explained in Section 2.2, but instead of starting from a fully masked sentence, it starts from a partially masked one. The student is optimized to match the teacher's soft labels and a temperature is used to control the smoothness of the labels. With enough capacity, the student is expected to imitate the behavior of $n$ consecutive steps of the teacher with merely one decoding pass.

## 3.1 Training Loss

We denote the output distribution after $n$ iterative steps on the partially masked sentence $\tilde{y}$ by $\mathcal{I}_\theta(\tilde{y}, x, n)$ where $\theta$ represents the parameters of the model. The distillation loss can be described as: $\sum_{i \in \xi(\tilde{y})} \text{KL}(p_{t,i} | p_{s,i})$ where $p_t = \mathcal{I}_{\theta_t}(\tilde{y}, x, n)$, $p_s = \mathcal{I}_{\theta_s}(\tilde{y}, x, 1)$ and $i$ in subscript denotes the index in the sentence. Note that the teacher's soft labels do not come from the same decoding iteration i.e. whenever a token is revealed, the corresponding soft labels are fixed in $p_t$. Thus, the student receives labels from various decoding steps of the teacher. Figure 2 depicts the process teachers follow to produce the labels for two iterative steps. From the student's point of view, the primary difference between DiMS and CMLM training (Section 2.1) is the use of soft labels generated by the teacher instead of the ground truth tokens.

To facilitate the distillation, we combine the KL-divergence with the Euclidean distance of the last layers' hidden states of the teacher and the student. This transfers the knowledge concealed within the hidden states that might not be discernible in soft labels. We refer to this as *hidden state loss*. Similar to the KL-divergence, the hidden state loss is computed over the masked indices. The fact that the hidden states have a positive impact on the distillation has also been observed in previous works (Romero et al., 2014; Sanh et al., 2019; Jiao et al., 2020).

To summarize, DiMS training loss has two terms: **i)** KL-divergence between distributions predicted by the teacher and the student. **ii)** The Euclidean distance between the last hidden states of two

---

**Algorithm 1** DiMS

---

**Require:** Data set $\mathcal{D}$, pre-trained model $\phi$, Hidden state loss factor $\lambda$, teacher steps $n$,
  EMA momentum $\mu$, learning rate $\eta$
  $\theta_t, \theta_s \leftarrow \phi$          ▷ Initialize teacher and student
  **while** not converged **do**
    $(\mathbf{x}, \mathbf{y}) \sim \mathcal{D}$          ▷ Sample data
    $\tilde{\mathbf{y}} \sim \mathbf{M}(\mathbf{y})$          ▷ Sample masking
    $\mathbf{p}_t \leftarrow \mathcal{I}_{\theta_t}(\mathbf{x}, \tilde{\mathbf{y}}, n)$          ▷ Run the teacher for $n$ iterative steps
    $\mathbf{p}_s \leftarrow \mathcal{I}_{\theta_s}(\mathbf{x}, \tilde{\mathbf{y}}, 1)$          ▷ Run the student for a single step
    $\mathcal{L}_{\text{DiMS}} \leftarrow \sum_i \text{KL}(\mathbf{p}_{t,i}|\mathbf{p}_{s,i}) + \lambda\|\mathbf{e}_{t,i} - \mathbf{e}_{s,i}\|^2$          ▷ Compute the DiMS loss
    $\theta_s \leftarrow \text{Optimizer}(\theta_s, \nabla_{\theta_s} \mathcal{L}_{\text{DiMS}}, \eta)$          ▷ Gradient based optimization of the student
    $\theta_t \leftarrow (1 - \mu)\theta_s + \mu\theta_t$          ▷ EMA Update of the teacher
  **end while**

---

models. Denoting teacher's and student's last hidden state by $\mathbf{e}_t$ and $\mathbf{e}_s$, DiMS loss can be written formally as:

$$\mathcal{L}_{\text{DiMS}} = \sum_i \text{KL}(\mathbf{p}_{t,i}|\mathbf{p}_{s,i}) + \lambda\|\mathbf{e}_{t,i} - \mathbf{e}_{s,i}\|^2$$
$$\text{where} \quad \mathbf{p}_t = \mathcal{I}_{\theta_t}(\tilde{\mathbf{y}}, \mathbf{x}, n) \quad \text{and} \quad \mathbf{p}_s = \mathcal{I}_{\theta_s}(\tilde{\mathbf{y}}, \mathbf{x}, 1).$$

The hyper-parameter $\lambda$ controls the contribution of hidden state loss. When the distillation is completed, the student is used for evaluation following the teacher's inference process explained in Section 2.2, thus the computation remains the same.

### 3.2 EMA Update of the Teacher

As the distillation progresses, the performance gap between multiple steps of the teacher and a single-pass of the student shrinks, making the teacher's labels less informative. Two approaches can be considered to sustain the usefulness of the teacher's labels: **i)** Increasing the number of teacher's iterative steps. **ii)** Restarting the distillation where the recently optimized student becomes the new teacher and repeating this process several times, i.e. $\theta_t^{(n)} \leftarrow \theta_s^{(n-1)}$. The former makes the training more expensive as the number of sequential steps grows and the latter requires repeated distillation rounds leading to a longer training time.

Instead, we propose updating the teacher with the student's recently learned knowledge. The idea is that as the student's single-step output approaches the teacher's multi-step, the student's multi-step performance would improve as well, and it is in the interest of the distillation process to use the improved student as the new teacher. However, replacing the teacher directly with the student would hurt the training stability and can lead to a pathological solution of mapping everything to a constant vector. This degenerate solution shortcuts the $\mathcal{L}_{\text{DiMS}}$ loss by setting it to a global minimum of zero. To alleviate this, we update the teacher with a slow-exponential-moving average of the student, which transfers the new knowledge learned by the student to the teacher in a controlled manner. The updated teacher now provides a better training target for the student, creating a positive feedback loop between the two models. Note that even with the slow-moving average, the degenerate solution is inevitable with long enough training. In practice, we find that it can be appropriately controlled by learning rate tuning and early stopping. The teacher can also benefit from the ensembling effects of the EMA (Izmailov et al., 2018). Figure 1 depicts DiMS training conceptually.

Overall, DiMS introduces three new hyper-parameters: **i)** n, the number of iterative steps of the teacher. **ii)** $\lambda$, the coefficient that controls the balance between KL-divergence and hidden state loss, and **iii)** $\mu$, the momentum of teacher's EMA update. Algorithm 1 contains the pseudo-code for DiMS training.

## 4 Experiments

### 4.1 Experimental Setup

We use the Fairseq library (Ott et al., 2019) for all the experiments and follow the default data splits. All models are transformers with encoder-decoder architecture, each having 6 layers and

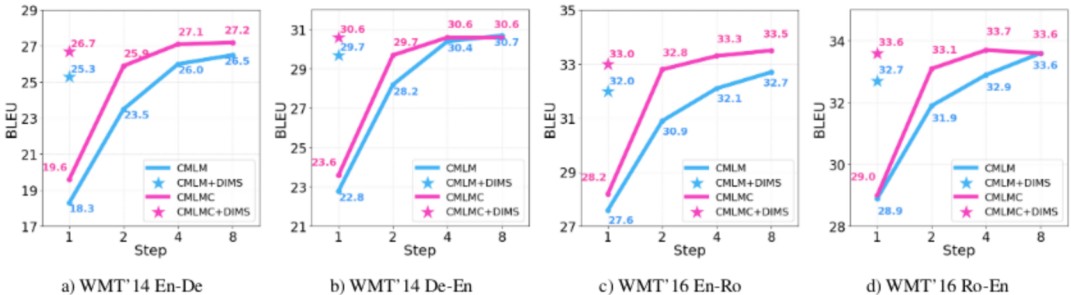

Figure 3: CMLM and CMLMC models distilled with DiMS on four WMT tasks. For each teacher we plot performance across various decoding steps and contrast it with a single-step performance of the student.

512-dimensional hidden states. Adam optimizer with inverse squared root learning rate scheduler is used along with mixed precision. EMA and hidden state loss are leveraged with two iterative steps of the teacher unless otherwise stated. We use early stopping based on single-step BLEU score on the validation set. The final model is the average of 5 best checkpoints. Dropout is disabled for the teacher and the student since empirical improvements are observed. We conduct experiments on both the raw and distilled dataset that is obtained from an autoregressive model (Gu et al., 2018). Training is done with 4 Tesla V100 GPUs (32 GB) and we report all the hyper-parameters in Section C of the appendix. The extra computational cost of distillation is a fraction of original training. We report a detailed comparison in Section F of the appendix.

## 4.2 MAIN RESULTS

Our main experiments are conducted on WMT'14 En-De and WMT'16 En-Ro datasets with two models: **i)** CMLM, a pivotal work in iNAT literature showing the effectiveness of conditional masked language models. **ii)** CMLMC, a recent work improving CMLM by incorporating a correction mechanism. The corresponding official repositories are used to train the teachers. Both models exploit a length predictor that is conditioned on the encoder's hidden states. To make the length predictor compatible with changes in the encoder, we keep the length predictor loss during distillation.

Figure 3 contrasts the single-step BLEU score of students with teachers evaluated for various number of decoding steps. DiMS considerably improves the translation quality of the single-step inference, reducing or eliminating the gap with multi-step inference. For example, on the WMT'14 De-En dataset, the single-step of CMLMC+DiMS matches the teacher's 8-step performance. We also compared our best single-step model with strong baselines in Table 1 showing the effectiveness of DiMS. Note that noisy parallel decoding (NPD) refers to the usage of the autoregressive model that generated the distilled the data to rerank the NAT's predictions with different sentence lengths. NPD is only applicable to models with length predictor.

The performance of the leading iNATs is at best similar to the autoregressive model used for sequence level knowledge distillation. This limits the final performance of iNATs and makes training without distillation desirable (Huang et al., 2021). The columns corresponding to raw dataset in Table 1 show that DiMS improves the raw performance by a large margin even more than the corresponding distilled variant. For instance, DiMS gets more than 12 BLEU scores improvements on single-step evaluation of CMLMC.

For one decoding pass, when raw variants of CMLMC are distilled with DiMS the performance is superior to training on the distilled dataset (without DiMS). This makes DiMS preferable to sequence-level knowledge distillation. Nevertheless, the best performance is obtained when the two distillation approaches are combined. On distilled datasets, DiMS either is competitive with state of the art or outperforms it. To the best our knowledge, DiMS achieves a new state of the art on single step evaluation over raw datasets.

It is not completely clear why knowledge distillation works in general. But when it comes to DiMS, we hypothesize that the labels generated by the teacher make the task simpler for the student. In other words, it is difficult for the model to close the gap between its single step prediction and ground truth while distillation with teacher-generated labels reduces this gap. The importance of the gap between labels and the model capacity has also been observed before Mirzadeh et al. (2020).

| Model | WMT'14 | | | | WMT'16 | | | |
|---|---|---|---|---|---|---|---|---|
| | En-De | | De-En | | En-Ro | | Ro-En | |
| | Raw | Dist. | Raw | Dist. | Raw | Dist. | Raw | Dist. |
| CTC (Libovický & Helcl (2018)) | 17.4 | - | 19.8 | - | 19.9 | - | 24.7 | - |
| CMLM (Ghazvininejad et al. (2019)) | 10.6 | 18.1 | - | 21.8 | 21.2 | 27.3 | - | 28.2 |
| SMART (Ghazvininejad et al. (2020b)) | - | 18.6 | - | 23.8 | - | - | - | - |
| CMLMC (Huang et al. (2021)) | 11.7 | 19.6 | 16.4 | 23.6 | 21.4 | 28.2 | 21.8 | 29.0 |
| Aux. Reg. (Wang et al. (2019)) | - | 20.7 | - | 24.8 | - | - | - | - |
| Bag-of-ngram (Shao et al. (2020)) | - | 20.9 | - | 24.6 | - | 28.3 | - | 29.3 |
| Hint-based Loss (Shao et al. (2020)) | - | 21.1 | - | 25.2 | - | - | - | - |
| Bigram CRF (Sun et al. (2019) | - | 23.4 | - | 27.2 | - | - | - | - |
| AXE (Ghazvininejad et al. (2020a)) | 20.4 | 23.5 | 24.9 | 27.9 | 30.5 | 30.8 | 31.4 | 31.5 |
| EM+ODD (Sun & Yang (2020) | - | 24.5 | - | 27.9 | - | - | - | - |
| ENGINE (Tu et al. (2020)) | - | - | - | 28.1 | - | - | - | 28.2 |
| Imputer (Saharia et al. (2020)) | 15.6 | 25.8 | - | 28.4 | - | 32.3 | - | 31.7 |
| OAXE (Du et al. (2021)) | 22.4 | 26.1 | 26.8 | 30.2 | - | 32.4 | - | 33.3 |
| FullyNAT + GLAT(Gu & Kong (2020)) | 21.8 | 27.2 | - | 31.4 | - | **33.7** | - | 34.2 |
| Flowseq + NPD (m=30) (Ma et al. (2019)) | 21.15 | 25.3 | 26.04 | 30.7 | 31.7 | 32.2 | 32.5 | 32.8 |
| GLAT + NPD (m=7) (Qian et al. (2021) | - | 26.5 | - | 31.0 | - | 32.9 | - | 33.5 |
| CMLMC + DiMS | 23.2 | 26.7 | 29.3 | 30.8 | 31.2 | 33.0 | 32.7 | 33.6 |
| CMLMC + DiMS + NPD (m=7) | **23.7** | **27.3** | **29.9** | **31.6** | **32.2** | **33.7** | **33.6** | **34.5** |

Table 1: Comparison of the single-step test set BLEU score with previously published works.

### 4.3 RESULTS ON AN ALIGNMENT BASED MODEL

To show the versatility of DiMS, we conduct experiment on alingment-based models leveraging Connectionist Temporal Classification (CTC) (Graves et al., 2006) objective. The main difference between alignment-based models (Libovický & Helcl, 2018) and CMLM family is the method used for modeling the output length. In an alignment-based model, output length is set to a large enough number (e.g. twice the length of the source sentence). By introducing a special blank token, the model has the ability to decide the sentence length since blanks are removed from the final prediction. Training such models requires an objective that considers all the alignments mapping to the same target which is what CTC objective does efficiently via dynamic programming. For example if the target sentence is "ABC" and the output length is set to four, then CTC considers "_ABC", "A_BC", "AB_C", "ABC_" as valid target sentences where "_" denotes the blank token.

Imputer (Saharia et al., 2020) is among a few iterative models in this family. There is no official implementation of Imputer available online, therefore we implement a version ourselves (denoted with †) [1]. We compare our implementation with the original paper in Section A of the appendix and show that two models have a similar performance. Table 2 summarizes the results of the base DiMS applied to Imputer for both directions of the WMT'14 English-German dataset. DiMS outperforms its teacher which shows that the applicability of DiMS is not limited to models trained with the cross-entropy objective. Further details of Imputer training and distillation are explained in Section D of the appendix.

| Method | WMT'14 En-De | WMT'14 De-En |
|---|---|---|
| Imputer† | 25.87 | 28.96 |
| Imputer† + DiMS | **26.43** | **29.75** |

Table 2: Sinlge-step test set BLEU score for Imputer models trained on WMT'14 English-German.

---

[1] Based on the following implementation: https://github.com/rosinality/imputer-pytorch

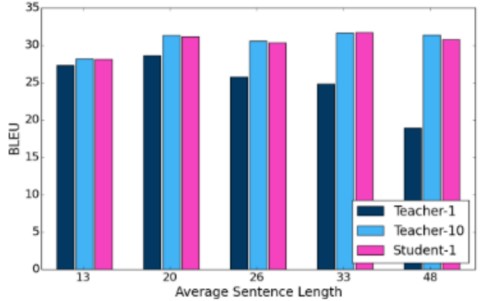

Figure 4: Test set BLEU score on WMT'14 De-En splitted based on the target sentence length for CMLM teacher and student.

Table 3: Single-step test set BLEU score for models trained with U-DiMS.

| Method | WMT'14 De-En |
|---|---|
| CMLM | 22.77 |
| CMLM + U-DiMS | 29.45 |
| CMLM + DiMS | **29.74** |
| CMLMC | 23.63 |
| CMLMC + U-DiMS | 30.52 |
| CMLMC + DiMS | **30.81** |

### 4.4 UNSUPERVISED DiMS

In previous sections, we assume access to a parallel dataset and feed a partially masked reference sentence to both student and teacher. One can use the teacher to generate synthetic target sentences during the distillation. This relaxes the dependence on the references and enables using monolingual datasets for distillation. As usual, there is a trade-off between computation and sample quality i.e. using more decoding passes leads to better data while increasing the computational requirements. We refer to this unsupervised distillation variant as U-DiMS. Note that unsupervised only refers to the distillation, and for training the teacher we still require access to a parallel dataset. The only distinction between U-DiMS and DiMS is the usage of synthetic data generated by the teacher and the remaining parts are untouched. We run U-DiMS on WMT'14 De-En for CMLM and CMLMC using two iterative steps to generate the synthetic samples. Table 3 shows the effectiveness of U-DiMS, obtaining a similar performance to DiMS.

### 4.5 ABLATION STUDIES

We conduct all of the ablation studies on CMLM over WMT'16 En-Ro as it is smaller than WMT'14 and the validation set is used for evaluation.

#### 4.5.1 HIDDEN STATE LOSS

To investigate the effects of hidden state loss, we conduct an ablation study in this section. The first block in Table 4 includes BLEU scores for the base DiMS model with and without this term. The single-step performance of the distilled model is improved over 2 BLEU points by leveraging this loss. This supports the fact that the hidden states contain extra information that is not available in soft labels. The exact value of $\lambda$ is selected based on a grid search reported in Section E of the appendix.

#### 4.5.2 EMA

In order to establish the computational advantages of the slow-moving average, we compare it with running the base variant for 9 iterative rounds. Figure 5 demonstrates that the EMA variant is able to match the iterative distillation with far fewer updates (almost equal to one round of the distillation).

We observed that it is essential to move the teacher toward the student slowly. For example, when $\mu \leq 0.9$, the collapse to a degenerate solution (explained in Section 3.2) occurs before the end of the first epoch. We plot the validation curve for various values of $\mu$ in Section B of the appendix showing the importance of the slow-moving average.

#### 4.5.3 TEACHER DECODING STEPS

One hyper-parameter in DiMS algorithm is the number of teacher's decoding steps. In order to investigate the effect of this hyper-parameter, we set it to 2, 4, and 8 while turning EMA on and off. The two bottom blocks of Table 4 include the results of this ablation. Although running the teacher for 4 decoding steps shows superior performance without EMA, as soon as we turn it on the gap disappears. This shows that EMA can gradually improve the teacher and remove the need for several

| Method | 1-Step BLEU |
|---|---|
| CMLM | 25.77 |
| CMLM + DiMS | 30.85 |
| CMLM + DiMS - Hidden. | 28.69 |
| CMLM + DiMS (T=4) | 31.04 |
| CMLM + DiMS (T=8) | 30.97 |
| CMLM + DiMS + EMA | **31.63** |
| CMLM + DiMS (T=4) + EMA | 31.52 |
| CMLM + DiMS (T=8) + EMA | 31.36 |

Table 4: BLEU score on WMT'16 En-Ro validation set with beam length set to one as its done for early stopping. T stands for the number of teacher decoding steps and it is set to two if not specified.

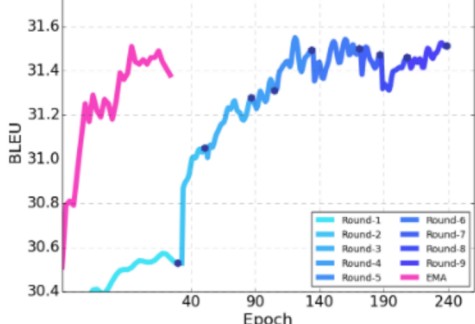

Figure 5: Validation set BLEU score on WMT'16 En-Ro for iterative distillation and a EMA model. Each round of iterative distillation is shown with a unique color and the end of the round is noted by a black dot. The number of steps differs in various rounds as we use early stopping.

iterative steps. Thus, we find no reason to set this hyper-parameter larger than 2 as it only increases distillation's computational cost.

### 4.6 ANALYSIS

We study the effect of target sentence lengths on DiMS performance. The test set is divided into five equally-sized buckets based on the target length. The BLEU scores are reported for each bucket in Figure 4. The main benefit of the iterative model is manifested by large sentences. The reason might be the fact that longer sentences require a context and modeling it becomes challenging with the conditional independence assumption in NAT. It is clear in Figure 4, that the performance is improved in every bucket. This improvement is most visible in the bucket with the highest average sentence length. This is because of the fact that the same bucket has the largest gap between the teacher's single and multi-step evaluation.

We combine the length predictor objective with ours to account for changes in the encoder's parameters. Interestingly enough, DiMS improves the performance of the length predictor as depicted in Figure 6. This shows that the encoder benefits from the distillation as well enhancing the length prediction accuracy.

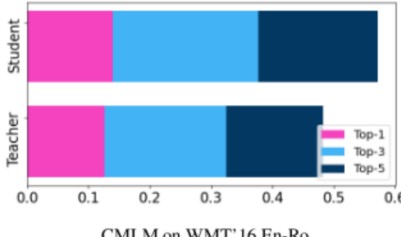
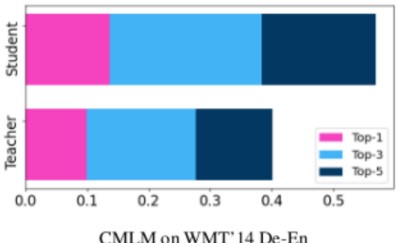

Figure 6: Comparison of teachers and students in predicting the target length. Top-1 means predicting the target exactly correct and Top-3 and Top-5 means being incorrect by 1 and 2 offsets, respectively.

Table 5 shows an example from the WMT'14 De-En dataset. The improvements in samples are evident by comparing the predictions of the teacher and the student with the target sentence. We provide more qualitative examples in Section G of the appendix.

| Target | The antibodies hunt down any nicotine molecules in the bloodstream , neutralising them before they reached the brain , preventing a smoker from getting a nicotine hit . |
|---|---|
| Teacher | The antibodies hunt the nicotine molecules molecblood neutralize them before reach brain a smoker not experience high nicotine . |
| Student | The antibodies hunt the nicotine molecules in the blood and neutralize them before they reach the brain , so a smoker does not experience a nicotine high . |

Table 5: A qualitative example from WMT'14 De-En along with teacher and student's predictions on CMLMC.

## 5 RELATED WORKS

Many techniques have been proposed for iterative non-autoregressive machine translation. Earlier attempts include denoising autoencoder (Lee et al., 2018) and insertion-deletion (Stern et al., 2019; Gu et al., 2019). More recently, Ghazvininejad et al. (2019) introduced the Mask-Predict improving the performance of iNATs by employing a conditional masked language model. CMLMC (Huang et al., 2021) and SMART (Ghazvininejad et al., 2020b) improve CMLM by incorporating a correction mechanism. DisCo (Kasai et al., 2020b) is another variant conditioning each token on an arbitrary subset of the other tokens. DiMS is entangled with the progress in this domain as it requires a pre-trained iterative teacher.

The ordering assumption in cross-entropy can make the NAT training challenging, therefore Ghazvininejad et al. (2020a) propose aligned cross-entropy (AXE), an objective that considers the best monotonic alignment between the target and the model's predictions. Du et al. (2021) relaxes the monotonic assumption and introduces Order Agnostic Cross-Entropy (OAXE). CTC (Libovický & Helcl, 2018) is a similar alignment-based objective that fixes the model output length and considers various alignments leading to the same target. Imputer (Saharia et al., 2020) extends CTC to benefit from iterative refinements.

GLAT (Qian et al., 2021) shows that the optimization challenges of iNATs can be mitigated by introducing a curriculum learning focusing on sentences with only a few masked tokens in the early stages of the training and gradually increasing the masking ratio. ENGINE (Tu et al., 2020) assumes access to a pre-trained autoregressive model and optimizes a NAT model to maximize the likelihood under the probability distribution defined by the pre-trained model.

Salimans & Ho (2021) applies a distillation technique similar to DiMS on generative models to decrease the number of required steps for generating high-quality images. In contrast to DiMS, the distillation is applied progressively. DiMS eliminates the need for progressive distillation by updating the teacher with EMA. Lastly, the proposed EMA has some resemblance to self-supervised learning techniques (Grill et al., 2020; Caron et al., 2021; He et al., 2020) where two models are updated, one through gradient-based optimization and the other one through EMA. Despite this similarity, the motivations are quite different. In self-supervised learning, EMA is proposed as a technique to remove large negative sets whereas here EMA enhances the quality of the labels generated by the teacher.

## 6 CONCLUSION

We introduce DiMS, an efficient distillation algorithm that enhances the translation quality of a pre-trained iterative model, especially with single-step translation. This is done by replicating the model's multi-step behavior through one decoding pass. The distillation can be repeated to achieve greater gains, but this increases the training time noticeably. We show that the same benefits are obtainable by setting the teacher as a moving average of the student while keeping the training time comparable to one round of the distillation. Experiments over raw and distilled datasets on four translation tasks with various models for supervised and unsupervised variants validate the effectiveness and versatility of DiMS.

Potential directions for future works include: **i)** The same family of iterative models have been applied to automatic speech recognition, thus DiMS is applicable to this domain. **ii)** One can combine a pyramid of techniques introduced for iNATs to obtain a strong iterative model and make it computationally efficient via DiMS. **iii)** Large monolingual sets can be used to distill models with U-DiMS.

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
