# OpenReview forum: "Speed Up Iterative Non-Autoregressive Transformers by Distilling Multiple Steps"
_ICLR.cc/2023/Conference — Submitted to ICLR 2023_

### Official Review · Reviewer_mVeK · 2022-10-23

**Confidence:** 5
**Correctness:** 3
**Technical Novelty And Significance:** 2
**Empirical Novelty And Significance:** 3
**Recommendation:** 5

**Clarity, Quality, Novelty And Reproducibility:**

This paper is generally well written and the experiments that have been conducted make sense and are useful for evaluating the proposed methods.

**Strength And Weaknesses:**

Pros:
1. An promising direction to improve fully NAT, which can maintain the advantage of decoding speed;

2. An interesting way to leverage the performance of multiple-step decoding via distillation.

Cons:

1. The approach is relatively complicated. It introduces 3 additional hyper-parameters (decoding steps of teacher model, interpolation weights of KL and Euclidean losses, and ), which potentially threat the robustness of the approach. Are these hyper-parameters the same across models (e.g. CMLMC and Imputer) and datasets (e.g. En-De and En-Ro).

2. The methods to distill multiple steps are practical but not innovative enough. The KL and Euclidean losses are straightforward, and widely-used in imitation learning.

3. The distillation pipeline still relies on an external teacher model (e.g. the pertained NAT model in this work), which prevents the model from learning from scratch (e.g. DAT (Huang et al., ICML 2022)).


Suggested Revisions:
1. The ultimate goal of NAT is  to become an independent translation model that achieves comparable with or even better performance than the AT models. Now the majority of NAT models are base setting (i.e., 6 layers of 512 dimensions) trained on medium-scale data (e.g., 4.5M WMT14 En-De), which still lags behind AT models (e.g., big or deep models trained on 20M WMT17 En-De data). It would be appreciated if the authors can validate the effectiveness of NAT models on big settings and/or larger datasets.

**Summary Of The Paper:**

Iterative NAT improves translation performance with multiple decoding steps, at the cost of sacrificing decoding speed. In response to this problem, this paper proposes to leverage distillation technique to imitate the behaviors of multiple steps with a single decoding step. To this end, the authors maintain two NAT models, namely teacher and student.

The student model learns to predict the multiple-step output of the teacher with two additional training objectives for the imitation learning: 1) a KL-divergence loss of the probability distributions of target tokens between the teacher and the student; 2) an Euclidean distance between the last layer representations of the two models.

As the training processes, the performance of student model is improved (for both single- and multiple- decoding steps), which can further enhance the teacher model. Accordingly, the teacher model is updated by the student via a slow-exponential-moving average.

Experimental results show that the proposed approach consistently improves performance over CMLMC (Huang et al., 2022) across language pairs on both distilled and raw data.

**Summary Of The Review:**

This paper focuses on improving the performance of single-step NAT (on raw data), which is a promising and important direction to make NAT be independent model. The proposed Distill Multiple Steps (DiMS) method is effective across datasets and model architectures.

However, the method introduces several hyper-parameters, which require a grid-search for individual dataset (e.g. Section 4.5). This method also relies on an pretrained teacher model, which prevents NAT from learning from scratch.

---

> ### Author Response · Authors · 2022-11-11
> **Response to Reviewer mVeK**
>
> We thank the reviewer for their feedback.
>
> - “...It introduces 3 additional hyper-parameters…which potentially threat the robustness…” while we introduce three hyper-paramters, the model is not too sensitive to them. For example, the number of decoding steps of the teacher n, is set to 2 in all the experiments except in the ablation study of section 4.5.3. These hyper-paramters are shared among various datasets which further shows the robustness of the algorithm to the hyper-parameters.
>
> - “distill multiple steps are practical but not innovative enough…”. We are not aware of any work in machine translation using distillation to decrease the number of decoding passes.
>
> - “The distillation pipeline still relies on an external teacher model which prevents the model from learning from scratch”. Training from scratch is another parallel interesting research direction. We believe both approaches are valuable and can help each other. Given any improved iNAT model such that its multi step translation quality is better than its single step translation, DiMS can be used to further improve its single step translation quality.
>
>
> - “...Now the majority of NAT models are base setting trained on medium-scale data, which still lags behind AT models. It would be appreciated if the authors can validate the effectiveness of NAT models on big settings and/or larger datasets.”
> The main reason for selecting the medium scale data is to make the comparison with the baseline models possible. Furthermore, we do not have access to large scale compute.

---

### Official Review · Reviewer_oFCe · 2022-10-25

**Confidence:** 4
**Correctness:** 4
**Technical Novelty And Significance:** 2
**Empirical Novelty And Significance:** 2
**Recommendation:** 5

**Clarity, Quality, Novelty And Reproducibility:**

The uploaded file seems like an image-converted pdf, making it difficult to read.

**Strength And Weaknesses:**

Strength: reducing the number of iterations for the non-autoregressive approach is a critical topic for the field, and the use of progressive distillation is a reasonable choice.
Weakness: the final results are worse than without applying the progressive distillation, and therefore making the contribution limited.

**Summary Of The Paper:**

The paper proposes to use progressive distillation for distilling non-autoregressive models, and update the teacher based on the exponential moving average of the student. The experiments assess the idea on the machine translation task and exhibit some improvement when comparing with single-step results.

**Summary Of The Review:**

While the idea of using progressive distillation for reducing the inference iterations of non-autoregressive models seems appealing, the resulting quality is not convincing and therefore the work provides limited contribution.

---

> ### Author Response · Authors · 2022-11-11
> **Response to Reviewer oFCe**
>
> We thank the reviewer for their feedback.
>
> - “...the final results are worse …” . The reviewer’s point is not clear to the authors as they already acknowledged that the student improved “… exhibit some improvement when comparing with single-step results”. The only goal of the paper is to improve single-step performance. Note that the focus of the recent non-autoregressive translation research is on single-step translation for reasons explained in the introduction.
>
> Figure 3 and Table 1 show that after applying distillation, the student outperforms the teacher by a significant margin on single decoding evaluation.

---

### Official Review · Reviewer_ABHG · 2022-10-25

**Confidence:** 4
**Clarity, Quality, Novelty And Reproducibility:** The contribution of this study is sma…
**Correctness:** 3
**Technical Novelty And Significance:** 2
**Empirical Novelty And Significance:** 2
**Recommendation:** 5

**Strength And Weaknesses:**

Strengths:

1. The training of the distilled student model obtained improved training efficiency with a certain translation quality.

2. The proposed approach gained improvement on the CMLM on the WMT14 EN-De and WMT16 Ro-EN benchmarks.

Weaknesses:

1. For many comparison methods in Table 1, the reported results appeared to be lower than the original paper (e.g., the CMLM model).

2. If the training time of the teacher model is also calculated, the efficiency of the final student model may not necessarily be improved.

3. As shown in Figure 3, the BLEU scores of the proposed method were inferior to those of the baseline CMLM and CMLMC models.

4. There were too many hyper-parameters (e.g., n, lambda, and p) in the proposed DiMS.

**Summary Of The Paper:**

This article proposed a distill multiple steps method to decrease the number of required steps to speed up iterative non-autoregressive transformer.

**Summary Of The Review:**

Please see review in the previous sections.

---

> ### Author Response · Authors · 2022-11-11
> **Response to Reviewer ABHG**
>
> We thank the reviewer for the feedback.
>
> -  “The training of the distilled student model obtained improved training efficiency with a certain translation quality”, We started with a pre-trained NAT model and our goal is not to improve the training efficiency. In fact, one epoch of the distillation process takes longer than one epoch of training as reported in section F in the appendix (although the number of necessary epochs is significantly lower). The main goal was to improve single step performance.
> - “... Table 1, the reported results appeared to be lower…”. As stated in the caption of Table 1, we report evaluation with one decoding pass of iNAT models whereas in their original papers they used up to 10 decoding steps. One step evaluation is aligned with the original motivation of the paper about reducing the latency and speeding up the translation.
> - “If the training time of the teacher model is also calculated, the efficiency of the final student model may not necessarily be improved.” The reason for referring to training time or efficiency for either student or the teacher is not clear as there is no claim toward training/distillation efficiency. The main benefit of DiMS comes at inference time. The only reason we provide the additional cost of distillation in appendix F is to show that the extra computation cost is not considerable and it is a fraction of the original training cost.
> - “… the BLEU scores of the proposed method were inferior to those of the baseline …”. In the distillation framework with a teacher and student, ideally the student matches the teacher's performance. However, empirically the student might fall behind the teacher. Note that the student outperforms the single step and two-step evaluation of the teacher significantly.
> - “... too many hyper-parameters …”. Our analysis shows that DiMS is not too sensitive to hyper-parameter values.
>
> We want to emphasize the fact that the main focus of DiMS is to decrease the latency of translation at evaluation time.

---

### Decision · Program_Chairs · 2023-01-20

**Decision:**

Reject

**Justification For Why Not Higher Score:**

N/A

**Justification For Why Not Lower Score:**

N/A

**Metareview: Summary, Strengths And Weaknesses:**

The paper proposes to use progressive distillation for distilling non-autoregressive models. While the reviewers admits the it is a promising direction to improve NAT, all the reviewers point out the quality of the experiments is not satisfying. Both in terms of the replication of the baselines and the setup for the proposed model. The paper needs to be further improved on these matters before it reaches the acceptance bar of ICLR.